# Psychological Distress in Patients Who Needed Invasive versus Non-Invasive Ventilation Following SARS-CoV-2 Viral Infection

**DOI:** 10.3390/brainsci14030189

**Published:** 2024-02-20

**Authors:** Laura Abbruzzese, Benedetta Basagni, Alessio Damora, Giulia Salti, Giulia Martinelli, Carmen Gambarelli, Alessandra Maietti, Cristiano Scarselli, Leonardo Carlucci, Pierluigi Zoccolotti, Mauro Mancuso

**Affiliations:** 1Tuscany Rehabilitation Clinic, Montevarchi (AR), 52025 Montevarchi, Italy; abbruzzese@crtspa.it (L.A.); damora@crtspa.it (A.D.); saltig9@gmail.com (G.S.); giulia.martinelli.fi@gmail.com (G.M.); scarselli@crtspa.it (C.S.); pierluigi.zoccolotti@uniroma1.it (P.Z.); 2Complex Structure of Rehabilitative Medicine, University-Hospital of Modena and Reggio Emilia, 41121 Modena, Italy; gambarelli.carmen@aou.mo.it; 3Specialist Rehabilitation Unit, Fondazione Poliambulanza Istituto Ospedaliero, 25133 Brescia, Italy; alessandra.maietti@poliambulanza.it; 4Department of Humanities, Letters, Cultural Heritage and Educational Studies, Foggia University, 71121 Foggia, Italy; leonardo.carlucci@unifg.it; 5Department of Psychology, Sapienza University of Rome, 00185 Rome, Italy; 6Recovery and Functional Re-Education UOC–Grosseto Area, USL South-East Tuscany, 58100 Grosseto, Italy; mauro.mancuso@uslsudest.toscana.it

**Keywords:** COVID-19, intensive care, psychological distress, anxiety, depression

## Abstract

The COVID-19 pandemic has affected the mental health of those who survived the illness but underwent long treatment and hospitalization. Much research has highlighted signs of emotional distress in those who experienced intensive care, and the procedures implemented to fight the infection. The present study investigated the effects of the illness experience in 40 subjects admitted to a rehabilitation unit after discharge from intensive care by focusing on the possibility of differences in emotional well-being depending on the type of ventilation. The results of the administration of psychological scales for anxiety, depression, and post-traumatic stress disorder showed that many subjects experienced some form of emotional distress. There were no differences between patients who underwent invasive ventilation and those who did not.

## 1. Introduction

The SARS-CoV-2 infection is a highly pathogenic and contagious disease that primarily presents with respiratory manifestations but that involves a range of symptoms [1]. During the pandemic, some patients required hospitalization in intensive care units for ventilatory support and, after discharge, some patients had to be admitted to multi-disciplinary rehabilitation services.

These long clinical pathways resulted in significant psychological distress, with frequent symptoms of depression, anxiety, and post-traumatic stress disorder (PTSD) [2,3]. Some studies have focused on the effects of ventilation procedures administered in intensive care units, the experience of emotional distress, and cognitive processing in COVID-19 patients [4,5].

Mechanical ventilation, also known as “invasive ventilation” (IV), supports patients with respiratory failure through tracheostomy or endotracheal tubes. “Non-invasive ventilation” (NIV) refers to respiratory support without direct tracheal intubation, using patient/ventilator interfaces in the form of facial masks. Therefore, it does not require deep sedation. Non-invasive ventilation is the first choice in treating respiratory failures and can effectively prevent intubation. Nevertheless, a lack of response to NIV may lead to IV.

The scientific literature has reported signs of psychological distress in patients receiving both types of ventilatory support. Chronic psychological distress and post-traumatic stress disorder have also been reported as complications of infection by other coronaviruses such as MERS [6] and SARS [7]. In fact, studies from the pre-COVID-19 era specifically compared the two types of ventilatory assistance and reported interesting results. In Shaw et al.’s study [8], subjects requiring invasive ventilation were significantly more likely to experience symptoms of PTSD (35%) and they reported higher levels of intrusion and avoidance symptoms compared to patients treated with non-invasive ventilation (10%), even though both groups were suffering from respiratory failure and both groups reported similar recollections of their hospital experiences. These results appear somewhat unexpected since patients undergoing invasive ventilation are sedated and cannot have a conscious memory of events.

However, one variable that could differently influence the subjective perception of distress connected with admission to intensive care could be the level of consciousness. In Volpato et al.’s study [9], the need for NIV increased fear and anxiety significantly in COVID-19 patients compared to the remission period and led to feelings of loss of control and decreased psychological well-being due to the uncertainty surrounding the disease in the acute setting. Patients who were aware of the critical situation created by COVID-19 could have been more conscious of the lack of contact with family and might have experienced great psychological stress. The latter increases pro-inflammatory markers and contributes to the increase in neuroinflammation and, therefore, increases symptoms of depression associated with sleep disorders, especially insomnia [10]. It could be hypothesized that deep sedation protects against the traumatic experience linked to awareness. Previous studies have hypothesized that PTSD symptoms, particularly avoidance and hyperarousal criteria, might occur in subjects who have no conscious recollection of their traumatic experiences due to implicit processing of the traumatic experience or internal reconstruction of memories from reports about their hospitalization [11,12,13].

In the COVID-19 population, differences were reported depending on the type of respiratory assistance in cognitive processing but not in emotional reactions at admission [4,5]; however, this pattern could be ascribed to the modulating effect of aging [4]. In a study by Alemanno et al. [4], a group of patients who had undergone invasive ventilation and sedation showed better cognitive performance; however, this group was younger than the other groups of patients who received non-invasive respiratory support in the acute phase of the disease. The effect of age on cognitive change has been well documented in the scientific literature [14]. Some cognitive abilities, such as vocabulary, are resilient to brain aging and may even improve with age. Other abilities, such as conceptual reasoning, memory, and processing speed decline gradually over time [15].

What might characterize the COVID-19 population from other patients hospitalized in intensive care for pathologies of other etiologies is the presence of a social dimension, in addition to the individual one, and the awareness of having a severely life-threatening infection. The pandemic and related restrictions affected people’s personal, occupational, and social lives in various ways and those who experienced the intensive care setting underwent the additional psychological burden of possible death in the absence of a caregiver by their side.

The present multicenter study investigated the psychological distress (specifically anxiety, depression, and symptoms of PTSD) of patients infected with COVID-19 to verify whether there were differences between patients who needed or did not need invasive ventilation, taking into account age, trait anxiety variables, and cognitive functioning at the time of assessment.

In agreement with previous studies on the non-COVID-19 population, we tested the hypothesis of higher rates of distress in IV compared to NIV within a probabilistic Bayesian framework.

## 2. Materials and Methods

### 2.1. Participants

We recruited a sample of 41 patients infected by COVID-19 admitted to the rehabilitation departments of the four Italian hospitals that participated in the study.

Inclusion criteria were (a) previous infection with SARS-CoV-2, with hospitalization and either invasive or non-invasive ventilation; (b) negativization of the swab at the time of recruitment; (c) stable SatO_2_ without respiratory assistance or no more than three L/min; and (d) age between 18 and 90 years. Patients with previous psychiatric or neurological diseases, pre-morbid general functioning, who obtained a Barthel Index (BI) score [16] below 90, or who used drugs or alcohol were excluded from the study.

We divided the sample into two subgroups. The non-invasive ventilation (NIV) subgroup included patients who benefited from oxygen therapy without invasive ventilation (i.e., Venturi masks or reservoir masks, CPAP); the invasive ventilation (IV) subgroup included patients who underwent invasive mechanical ventilation (via an endotracheal tube or a tracheostomy tube).

### 2.2. Design and Procedure

Once dismissed from intensive care units, patients were examined and admitted to rehabilitation centers. When they were admitted to the rehabilitation unit, socio-demographic and clinical information was collected. All patients were submitted to a bedside assessment to define their psychological distress, and to a cognitive screening. Additional scales were also adopted to mitigate the influence of confounding factors, such as symptoms of anxiety in the pre-morbid period, age, or the effects of impaired cognitive functioning. The neuropsychological assessments were conducted by a clinical neuropsychologist. The functional scales were filled out by a medical doctor.

The study was approved by the Local Ethics Committee of the regional health service, i.e., the Tuscany South-East area, on 28 June 2021 (Protocol number: 202/000189; Project identification code: 19755), and the study was carried out according to the Declaration of Helsinki. Patients gave their written informed consent to participate in the study.

### 2.3. Psychological Assessment

The State–Trait Anxiety Inventory Form Y (STAI-Y) [17,18] is a self-report questionnaire commonly used in clinical settings that measures trait and state anxiety. The STAI-Y has 20 items that assess state anxiety (STAI-S) and 20 that assess trait anxiety (STAI-T). Items are rated on a 4-point Likert scale that ranges from “almost never” to “almost always”. State anxiety indicates how anxious the person feels “right at that moment”, while trait anxiety refers to a more lasting and stable condition of the personality that characterizes the individual continuously, regardless of a particular situation. This test was included to differentiate between patients with underlying anxiety traits, regardless of their condition at the time of assessment, and those who only exhibited symptoms of anxiety when they were hospitalized. The total score for both scales ranges from 20 to 80, with higher scores indicating more severe anxiety. Internal consistency varies between 0.91 and 0.95 for the STAI-S and between 0.85 and 0.90 for the STAI-T. The test–retest reliability (at a distance of one month) is 0.49 for the STAI-S and 0.82 for the STAI-T [18].

The Beck Depression Inventory-II (BDI-II) [19,20] is a 21-item self-report inventory designed to measure depression in adolescents and adults. Each item is rated on a 4-point Likert-type scale ranging from 0 to 3 and is based on the severity over the last two weeks (range: 0–63). The validation study of the BDI-II revealed excellent psychometric characteristics of the scale with an internal consistency of 0.92 and a test–retest correlation of 0.93 [19,20].

The COVID-19 Post-Traumatic Stress Disorder Questionnaire (COVID-19 PTSDQ) [21] is a self-report tool that is designed to assess post-traumatic stress disorder (PTSD) symptoms related to the COVID-19 pandemic and validated in the Italian population. The COVID-19-PTSD questionnaire was developed starting from the PTSD Check List for DSM-5 (PCL-5) questionnaire [22] and is focused on direct (e.g., fear of being infected) and indirect (e.g., social isolation) stress factors consequent to the COVID-19 emergency. It includes 19 items, each of which requires a response on a 5-point Likert scale, from 0 (not at all) to 4 (extremely) for a total score ranging from 0 to 76. The authors proposed a seven-factor structure (intrusion symptoms, avoidance symptoms, negative changes in thinking and mood, anhedonia, dysphoric arousal symptoms, anxious arousal symptoms, and externalizing behaviors) which has proven to fit well with the data. Scores above 1.5 standard deviations from the sample mean are indicative of more PTSD symptoms. A cut-off score of 26 was considered the best predictor of PTSD symptomatology, with a sensitivity of 0.91 and a specificity of 0.92 [21]. The internal consistency of the COVID-19 PTSD items was 0.94 and correlation coefficients for each item of the subscales were between 0.52 and 0.85 for the seven-factor model [21].

### 2.4. Neuropsychological Assessment

The Oxford Cognitive Screen (OCS) [23,24] is a screening tool for quickly assessing the patient’s cognitive functioning. It is structured around five cognitive domains: attention and executive functions, language, memory, number processing, and praxis. Visual field efficiency is also assessed with the Visual Field subtest (range: 0–4). In the language domain, the subtests are Picture naming (min–ma possible range: 0–4), Sentence reading (range: 0–15), and Semantics (range: 0–3). The subtests for the attention and executive functions domain are the Trails subtest (range: −13/+12) and the Broken hearts cancellation subtest, which provides three different sub-scores: whole hearts cancellation (i.e., the total number of complete hearts cancelled as a measure of selective visual attention; range: 0–50), space asymmetry (the difference between complete hearts cancelled in the left and right portions of the page as a measure of egocentric neglect; range: −20/+20), and object asymmetry (the difference between left and right broken hearts as a measure of allocentric neglect; range: −50/+50). The subtests for numerical cognition include a subtest of Number writing (range: 0–3) and Calculation (range: 0–4). The Praxis domain includes the Imitation subtest (range: 0–12). The subtests for the memory domain include the Orientation (range: 0–4), Recall and recognition (range: 0–4), and Episodic memory (range: 0–4) subtests. A cut-off score for each subtest is provided [24].

### 2.5. Statistical Analyses

First, we analyzed the percentage of deviant performance on each test. We used the transformation of the STAI-T raw scores into z-scores to exclude patients with pre-morbid stable conditions of anxiety. Critical z-score values when using a 95% confidence level are 1.96 standard deviations from the mean [25]. For the STAI-S scale, we used the classifications “no or low anxiety” (20–37), “moderate anxiety” (38–44), and “high anxiety” (45–80) to calculate the percentage of abnormal performance [18]. For the BDI-II scale, the raw scores were converted into percentile scores by considering a percentile below 85 as indicative of a lack of depression [19,20]. A cut-off score of 26 was used to calculate the percentage of patients who performed above the cut-off on the COVID-19 PTSDQ [21].

Second, we checked the normality of the data distribution using the Shapiro–Wilks test for small sample size (N < 50). Due to the unequal sample sizes of the IV and NIV groups, and to address the statistical power issue (i.e., the ability to detect a difference will diminish as the group sizes become more unequal), we performed a Bayesian Mann–Whitney U. Bayes factor (BF) as default BF01 that assesses how much the observed data relatively support the null hypothesis (H0) over the alternative hypothesis (H1). Larger values of BF01 indicate more support for H0 (e.g., a BF_01_ = 2 means that, given the data, the null hypothesis is two times more likely than the alternative hypothesis) [26]. Bayes factors range from 0 to ∞, and a BF value of 1 indicates that both hypotheses predicted the data equally well. Analyses were carried out on raw data with JASP 0.18.01 software (JASP Team, 2024).

## 3. Results

We enrolled 41 patients (Mean age = 64.3 ± 13.4; range = 19–81; 28 males and 12 females; mean education = 11 ± 3.4). The IV group comprised 25 males and 6 females (mean age = 64.3 ± 13.6; range = 19–80). The NIV group comprised 3 males and 6 females (mean age = 64.4 ± 13.5; range = 39–81). Only one was pathological on the trait anxiety scale of STAI and was excluded from the sample (z-score = 2.11). In this way, we limited the impact of factors related to pre-morbid traits that were unrelated to the experience of the illness and the hospitalization. Thus, the final experimental sample included 40 subjects. The demographic characteristics of the total sample and the two subgroups are presented in Table 1.

As regards neuropsychological functioning, 47.5% of the patients in the total group performed above the cut-off in the attention domain, 45.2% in the IV, and 55.6% in the NIV group. Memory and language domains were both compromised in 37.5% of cases. A 44.4% share of patients in the NIV group performed pathologically in the numerical cognition subtest, and 19.4% of the IV group failed in the Number writing and Calculation subtests. In the language and memory domains, the ventilated patients were compromised in 41.9% of cases, while in the NIV group, language, memory, and praxis were equally compromised (22.2%). The neuropsychological characteristics of the total sample and the two subgroups are presented in Table 2.

The data regarding psychological distress are presented in Table 3. On the Beck Depression Inventory, 35.5% of the patients in the IV subgroup and 33.3% in the NIV subgroup had mild to moderate depression. On the STAI scale, 48.3% of those in the IV subgroup had mild to high state anxiety, while 33.3% from the NIV group had mild to moderate state anxiety. COVID-19-related PTSD symptoms were 45.2% and 55.6% for the IV and NIV subgroups, respectively.

Due to the small sample size, we determined the distribution of the variables in the study to choose an appropriate statistical method. The Shapiro–Wilk test showed that the distribution of variables departed significantly from normality (S-W_range_ = 0.915–0.147, *p*-value < 0.01). Consequently, a Bayesian Mann–Whitney U test (labelled as W) with 10,000 seeds was computed for both neuropsychological and psychological sample characteristics (see Table 2 and Table 3). Unexpectedly, BF_01_ for all the variables in the analysis was ≥2, suggesting weak to moderate evidence for the H0 (no differences between the two groups) with a BF_01_ ranging from 1.74 to 3.02, except for the BI at admission variable where observed data largely supported 0.18 times more for the alternative hypothesis (i.e., there was a mean difference on the Barthel Index between the two groups) than for the null hypothesis (i.e., there was no mean difference on the Barthel Index between the two groups). Since the variance was equal to 0 after grouping one group, it was impossible to compute the U test for the Semantics, Visual field, and Number writing OCS domains.

Since the Bayes factor quantifies the relative predictive performance of two rival hypotheses [27], in a confirmative fashion we recomputed the same analysis by changing the Bayes Factor (BF_10_; H1 over H0) and formulating alternative hypotheses (i.e., IV > NIV). To sum up, these analyses confirmed the results reported above which indicate that there is weak evidence of group differences in the neuropsychological and psychological characteristics of the sample.

## 4. Discussion

In this study, we analyzed the psychological distress experienced by 40 patients who had been infected by COVID-19 to verify whether there were any differences between patients who had or who had not needed invasive ventilation. We found that some symptoms, such as anxiety and depression, as well as PTSD, were present but not substantial. These data are in keeping with previous observations of COVID-19 patients that reported both a high incidence of psychiatric symptoms and the activation of mechanisms of resilience and a positive reaction to the unfavorable event in these patients [2,3,28]. As expected, clinical derangement in ADL, measured by the Barthel Index, was more pronounced in patients who needed IV. However, the emotional well-being of the IV and NIV subgroups was similar. Furthermore, there were no differences in the cognitive processes of the two subgroups, as assessed by the OCS scale.

These results are at variance with those of studies on the non-Covid population, indicating the prevalence of distress in patients who required IV (e.g., [8]) but consistent with previous evidence on COVID-19 patients, which indicated a limited effect of IV on affective factors [4,5]. Furthermore, our results also did not show significant differences between groups with regard to cognitive functioning, which should be considered as inclusive of both pre-morbid aspects and possible effects of the COVID-19 infection. We also excluded one patient who showed trait anxiety on the STAI-T test to ensure that our sample was uninfluenced as much as possible by other potentially interfering variables. At any rate, the overall sample available for our study (particularly the sub-sample of subjects in the IV subgroup) was comparatively small and this constitutes a relevant limitation of the current study. Furthermore, the type of viral infection associated with SARS-CoV-2 has changed considerably and fortunately very few patients now require hospitalization with invasive or non-invasive ventilation, making it difficult to increase the sample size.

There is considerable evidence that during the pandemic the general population also suffered from emotional problems (e.g., ref. [29] for a review and a meta-analysis), such as anxiety, depression, and psychological distress. At that time, there was a collective awareness of a tragic experience with potentially devastating outcomes. Our results, together with the previous findings from COVID-19 patients hospitalized in intensive care units, indicate that the emotional reactions of these patients could have been less affected by the type of ventilation and, presumably, more by the perception of the high-life risk associated with the pandemic.

### Limitations

Although the present study has some limitations, its strength lies in the unique context of the ongoing pandemic. The greatest limitation concerns the unbalanced group sizes, with more subjects in the invasive ventilation group, which has an impact on the robustness of the statistical analysis. The study was conducted within a confined timeframe and marked by circumstances that were specific, non-repeatable, and beyond experimental control. The evolving nature of the pandemic precluded any possibility of increasing or balancing the sample size. Consequently, the results cannot be broadly generalized to other situations, aligning the study within the realm of practice-oriented research. This framework underscores the opportunity to study acute medical care in conditions that, as previously mentioned, are not subject to experimental manipulation for obvious ethical reasons [30].

## 5. Conclusions

Emotional distress (symptoms of anxiety, depression, and PTSD) is a frequent outcome of the subjective experience of hospitalization in intensive care and respiratory support maneuvers. The population affected by COVID-19 showed no significant differences based on the type of ventilation. There were some symptoms of anxiety and depression (as well as signs of PTSD) that indicated both a high incidence of psychiatric symptoms in patients infected with COVID-19 and the possible role of mechanisms of resilience and a positive reaction to the disturbing event.

## Figures and Tables

**Table 1 brainsci-14-00189-t001:** Demographic characteristics of the sample.

	All (N = 40)				IV (N = 31)				NIV (N = 9)					
			95% CI				95% CI				95% CI		BF_01_	W
	M	SD	Lower	Upper	M	SD	Lower	Upper	M	SD	Lower	Upper		
Education	11.0	3.4	10.0	12.1	10.6	3.2	9.5	11.8	12.4	4.3	9.2	15.7	2.06	100.0
Age	64.3	13.4	60.2	68.5	64.3	13.6	59.3	69.3	64.4	13.5	54.0	74.8	3.02	139.0
Time from onset (days)	95.7	44.2	82.0	109.4	98.3	45.4	81.7	115.0	86.9	41.0	55.4	118.4	2.91	158.5
BI at admission	36.2	36.3	25.0	47.5	23.5	27.8	13.3	33.7	80.0	27.3	59.0	101.0	0.18	27.0

Notes. BI: Barthel Index; IV: Invasive Ventilation group; NIV: Non-Invasive Ventilation group; CI: Credible Interval; a = the variance is equal to 0 after grouping on group; BF_01_: Bayes Factor H0 over H1; W = Mann–Whitney U test.

**Table 2 brainsci-14-00189-t002:** Neuropsychological characteristics of the sample.

	All (N = 40)					IV (N = 31)					NIV (N = 9)						
			95% CI		% Below Cut-Off			95% CI		% Below Cut-Off			95% CI		% Below Cut-Off	BF_01_	W
	M	SD	Lower	Upper		M	SD	Lower	Upper		M	SD	Lower	Upper			
**Oxford Cognitive Screen**																	
**Language**					37.5					41.9					22.2		
Picture naming	3.4	0.8	3.2	3.7	25	3.3	0.8	3.0	3.7	29	3.8	0.4	3.4	4.1	11.1	2.01	102.5
Semantics	3.0	0.0	3.0	3.0	0	3.0	0.0	3.0	3.0	0	3.0	0.0	3.0	3.0	0	a	a
Sentence reading	14.5	2.1	13.9	15.2	15	14.4	2.4	13.5	15.3	16.1	14.9	0.3	14.6	15.1	11.1	2.55	131.0
**Attention and Executive Functions**					47.5					45.2					55.6		
Whole hearts	43.0	7.6	40.6	45.4	45	42.7	8.3	39.6	45.9	41.9	43.8	5.2	39.7	47.8	55.6	2.91	136.0
Space asymmetry	0.1	3.6	−1.0	1.3	17.5	0.5	3.9	−1.0	1.9	19.4	−1.0	2.4	−2.8	0.8	11.1	1.74	169.0
Object asymmetry	−0.03	1.0	−0.3	0.2	5	0.0	1.0	−0.4	0.4	3.2	−0.1	1.2	−1.0	0.8	11.1	2.60	123.0
Trails Task																	
Baseline score	11.6	0.9	11.3	11.9	10	11.5	1.0	11.2	11.9	9.7	11.8	0.7	11.3	12.3	11.1	2.48	121.0
Shifting score	0.8	2.9	−0.1	1.7	12.5	0.9	3.1	−0.2	2.0	12.9	0.4	2.2	−1.3	2.2	11.1	2.78	144.5
**Numerical Cognition**					25					19.4					44.4		
Number writing	3.0	0.1	2.9	3.0	2.5	3.0	0.2	2.9	3.0	3.2	3.0	0.0	3.0	3.0	0	a	a
Calculation task	3.7	0.4	3.6	3.9	25	3.8	0.4	3.6	3.9	19.4	3.5	0.5	3.1	4.0	44.4	2.05	174.5
**Praxis**					20					19.4					22.2		
Imitating meaningless gestures	10.3	2.2	9.6	11.0	20	10.3	2.4	9.4	11.2	19.4	10.1	1.7	8.8	11.4	22.2	2.28	158.5
**Memory**					37.5					41.9					22.2		
Orientation	3.9	0.3	3.8	4.0	7.5	3.9	0.2	3.8	4.0	6.5	3.9	0.3	3.6	4.1	11.1	2.46	146.0
Recall and recognition	2.8	1.2	2.4	3.1	22.5	2.8	1.2	2.3	3.3	22.6	2.7	1.1	1.8	3.5	22.2	2.61	155.0
Episodic memory	3.7	0.6	3.5	3.9	22.5	3.6	0.7	3.4	3.9	25.8	3.9	0.3	3.6	4.1	11.1	2.32	118.0
**Visual field**	4.0	0.1	3.9	4.0	2.5	4.0	0.2	3.9	4.0	3.2	4.0	0.0	4.0	4.0	0	a	a

Notes. IV: Invasive Ventilation group; NIV: Non-Invasive Ventilation group; CI: Credible Interval; a = the variance is equal to 0 after grouping on group; BF_01_: Bayes Factor H0 over H1; W = Mann–Whitney U test.

**Table 3 brainsci-14-00189-t003:** Results of psychological scales in the whole sample and divided into the IV and NIV subgroups.

							IV (N = 31)				NIV (N = 9)						
			95% CI		%			95% CI		%			95% CI		%		
	M	SD	Lower	Upper		M	SD	Lower	Upper		M	SD	Lower	Upper		BF_01_	W
STAI-State	40.3	13.0	36.3	44.3	45	10.0	6.0	7.8	12.2	48.4	13.3	13.3	3.1	23.6	33.3	0.37	149.0
BDI-II-Total	10.8	8.1	8.2	13.3	35	7.9	4.5	6.2	9.5	35.5	10.2	9.4	3.0	17.4	33.3	0.38	140.5
BDI-II-Somatic-affective	8.4	5.9	6.6	10.2	47.5	2.1	2.0	1.4	2.9	51.6	3.1	4.2	−0.1	6.4	33.3	0.37	148.0
BDI-II-Cognitive	2.3	2.6	1.5	3.2	30	40.3	12.1	35.9	44.8	29	40.1	16.6	27.3	52.9	33.3	0.34	152.5
COVID-PTSDQ	23.8	14.1	19.4	28.2	47.5	23.9	14.1	18.7	29.0	45.2	23.5	14.8	12.2	34.9	55.6	0.36	140.5

Note. %: subjects with a deviant performance (see text for more details on the cut-offs for the different psychological scales); STAI: State–Trait Anxiety Inventory Form Y; BDI-II: Beck Depression Inventory-II; COVID-19 PTSDQ: COVID-19 Post-Traumatic Stress Disorder Questionnaire; M: males; F: females; IV: Invasive Ventilation group; NIV: Non-Invasive Ventilation group; CI: Credible Interval; BF_01_: Bayes Factor H0 over H1; W = Mann–Whitney U test.

## Data Availability

The raw data supporting the conclusions of this article will be made available by the authors on request.

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
