# Peer review of "Psychological Distress in Patients Who Needed Invasive versus Non-Invasive Ventilation Following SARS-CoV-2 Viral Infection"

_brainsci, 2024, doi:10.3390/brainsci14030189_

Round 1
Reviewer 1 Report
Comments and Suggestions for Authors
I attach a word file with my detailed comments and suggestions.

Author Response
Please find attached the response to the Reviewer

Reviewer 2 Report
Comments and Suggestions for Authors
The unique criteria of invasive or noninvasive ventilation on which is based this study is not sufficient relevant to estimate the psychological state of former patients.
Others criteria: reliability and validity.
Comments on the Quality of English Language
English quality is satisfying !
Author Response

(The authors gave the same response as above.)

Reviewer 3 Report
Comments and Suggestions for Authors
Thank you for inviting me to review this brief report. I found it very interesting. The authors tried to establish if different modalities of ventilation during COVID-19 infection lead to different psychological profiles. The found non-significant differences for all measures of interest. However, I have to raise severe methodological issues and do not find the presented methods and analyses sufficiently described or appropriate. I have to reject the manuscript in its current form. In the frequentist Null-hypothesis testing framework, evidence in favor of the Null hypothesis (in this case establishing that means are equal/do not "significantly" differ) cannot be established with a p-value. A non-significant p-value cannot be interpreted in any form. If the authors wish to establish evidence in favor of the Null hypothesis, they should either conduct equivalence tests or, preferably, conduct their analyses in a Bayesian framework.
l. 62: „extremely“ should be substituted with “severely” or some word alike. It sounds rather sensational and not scientific.
Section 2.3: Are there any information on the reliability of your measures? Since your study is centered on interpreting these instruments, it is crucial to establish their reliability both in general (e.g., meta-analyses of major studies) and your specific estimates of reliability (e.g., McDonald’s Omega).
ll. 118 – 136: What is the range for the COVID-19 PTSDQ and the OCS?
ll. 126 – 127: What does “high sensitivity and […] specificity” mean? Do you have concrete estimates for this assumption? If so, the reader might be highly interested to know these.
Table 1: What does “Pathological performance” mean? I don’t think that this was explained in the statistical analyses section. If this means that an individual crossed a certain cutoff, this must be defined in the methods section and be based on the relevant literature.
Table 2: This table seems rather confusing and is bloated with information that is maybe unnecessary? I don’t know because much of what is shown was not introduced earlier. What is the “Standardization sample”? Why was a z-score calculated? And how? What does it mean in this context? z-scores are normally distributed with mean 0 and SD 1, which is what you have shown for nearly all outcomes, but for what purpose?
Tables: For group comparisons, please report adequate effect sizes with confidence intervals for those group differences. A p-value is never a sufficient summary statistic.
ll. 181 – 182: Where is this analysis reported?
ll. 187 – 190: I am afraid that you cannot make those statements with your analysis plan at hand. A non-significant p-value of p > .05 does not indicate a "missing" difference. Your study could just as well be severely underpowered, which is likely to be the case given your sample size of N = 40 relevant individuals incorporated in the analyses. A non-significant p-value does not lead to a statement like “There was no difference”. There is a small difference, that frequentist null hypothesis testing just not deemed surprising enough if the Null hypothesis was true (https://doi.org/10.1053/j.seminhematol.2008.04.003). With a p-value, you cannot measure evidence in favor of the Null hypothesis.
ll. 172 – 174: With all due respect, but this statement is completely out of the frame. E.g., your BDI sum scores are rather normative and only indicate a mean light depressive syndrome for the NIV group. This is a rather questionable claim and approach to summarize your findings that lack any findings whatsoever (with the exception of significant BI score differences). Given your methods section, in which cutoff scores, on which you conclude your quite dubious assumptions, are not reported, this is simply sloppy – intentional or not -- and puts the whole manuscript itself at question.
Author Response

(The authors gave the same response as above.)

Reviewer 4 Report
Comments and Suggestions for Authors
Psychological Profile of Patients Hospitalized in a Rehabilitation Unit Following SARS-CoV-2 Viral Infection
The study addresses an interesting and important research question I was looking forward to seeing the results, especially when I read in the introduction that this was a multi-center study. Unfortunately, the methodology section reveals some rather important weaknesses in both the study design and the way it was conducted.
My main concerns are: extremely small sample size (n=40), and extremely uneven subsample distribution (31 vs. 9). I understand that fewer patients needed the invasive ventilation, but this type of size discrepancy makes statistical comparisons unreliable. There simply was not enough statistical power to detect any meaningful differences, even if they were some. I would strongly suggest the authors to try and recruit more patients for the NIV group – as it is, any use of parametric tests is questionable.
It is obvious that the needed sample size was not calculated a-priori, but the authors could calculate (using G*Power, for example) how much statistical power they in fact had with this sample size, considering the effect sizes.
Furthermore, a more detailed description of the sample is necessary: how many men, and how many women in each group? (This information would be crucial, even if it was not well known that women tend to score higher on both BDI and STAI).
The sample size should be reported earlier in the text (in abstract, and then in the participants section: as it is, the reader has to find table 1 to find this information).
Tests: Please provide psychometric characteristics for all the questionnaires (STAI, BDI, COVID-19 PTSDQ) on your sample (at least the reliability9. For COVID-19 PTSDQ scale also provide the information regarding the factorial structure and a more precise description: how many items in total? How many in each subscale? The cut-off score of 26 was across all items? Was one score computed or 7? Too many missing pieces of information.
Tables: both tables are scattered and hard to follow and lack appropriate descriptions. The OCS tests should be grouped by the cognitive ability they asses (add a column with label), the p values can never be 0 (irrespectively of the fact that SPSS output shows a zero, use the “p<.001” expression. What is F in table 2? It usually denotes an ANOVA, but here we have t-test presented in the last column, so Fs are either redundant or refer to something else, no described adequately.
Where did the STAI trait score disappear? The authors claim they used it as a control variable, but it is not reported in the result section at all.
Furthermore, the authors keep claiming they have controlled for cognitive profile, which doesn’t seem to be the case: if the cognitive tests were applied after the COVID episode (and that is what the method section states) then these variables were not controlled for, as all of these cognitive functions are easily influenced by health status, and the only way to control for them would be to have information regarding the cognitive functioning before the COVID infection and hospitalization. It is very likely that both groups had lower performance across numerous cognitive tasks following the infection (as this pattern has been documented on large samples in the last several years).
Comments on the Quality of English Language
English is satisfactory.
Author Response

(The authors gave the same response as above.)

Round 2
Reviewer 1 Report
Comments and Suggestions for Authors
A file is attached describing my final, mostly minor comments.

As noted in my review report, I felt that there were still many sentences with suboptimal wording, incorrect grammar, lengthy sentences, or sentences where the language impedes understanding of the core message. I suggest using an English language editing service.
Reviewer 2 Report
Comments and Suggestions for Authors
I d'ont ask another revision!
Author Response
We thank the Reviewer 2 for his/her contribution to the review process.
Reviewer 3 Report
Comments and Suggestions for Authors
This is the second review round for this manuscript. I highly appreciate the authors' effort to use statistical methods that can deliver answers to their hypotheses. Well done! This is a sign of constructive and collaborative scientific work.
I do only have a small addition. Insetad of reporting the BF10 (measuring the evidence for the alternative hypothesis in relation to the Null hypothesis), I would suggest using the BF01, which is the reciprocal of the BF10 (BF01 = 1 / BF10). In this case, the Bayes factor measures the evidence in favour of the Null hypothesis, which is what the authors try to establish here. So instead of reporting BF10 = 0.333 you could report BF01 = 3 which means that, given the data, the Nully hypothesis is three times more likely than the alternative hypothesis.
Otherwise I am satisfied with the manuscript as is.
Author Response
Thank you for your positive feedback and appreciation for our efforts in employing statistical methods to address our hypotheses. We are pleased to hear that you consider our work as a manifestation of constructive and collaborative scientific endeavor. We have modified the text as requested.
Reviewer 4 Report
Comments and Suggestions for Authors
Thank you for addressing my comments.
Also, I find the new version of data analysis (the Bayesian approach) much more suitable for your dataset than the earlier version.
Comments on the Quality of English LanguageEnglish is fine.
Author Response
We thank you for your feedback and your acknowledgement of our efforts to address your comments.